# Lignin Degradation via Chlorine Dioxide at Room Temperature: Chemical Groups and Structural Characterization

**DOI:** 10.3390/ijms24021479

**Published:** 2023-01-12

**Authors:** Shuxian Weng, Guixin Zhang, Yun Hu, Caiying Bo, Fei Song, Guodong Feng, Lihong Hu, Yonghong Zhou, Puyou Jia

**Affiliations:** 1Co-Innovation Center of Efficient Processing and Utilization of Forest Resources, Key Laboratory of Biomass Energy and Materials, Institute of Chemical Industry of Forest Products, Chinese Academy of Forestry (CAF), Nanjing 210042, China; 2College of Chemical Engineering, Nanjing Forestry University, Nanjing 210037, China

**Keywords:** chlorine dioxide, lignin, room temperature, degradation

## Abstract

Lignin degradation is an effective means of achieving the high-value application of lignin, but degradation usually requires the use of high temperatures and harsh reaction-conditions. This study describes a green, mild approach for the degradation of lignin, in which chlorine dioxide (ClO_2_) was used for the oxidative degradation of lignin (IL) in an acidic aqueous suspension at room temperature. The optimal process conditions were: 30 mL of ClO_2_ solution (2.5 mg·L^−1^), pH 4.5 and 3 h. The FT-IR, NMR (^1^H NMR, 2D-HSQC and ^31^P NMR), XPS and GPC analyses indicated that lignin could be degraded by ClO_2_ relatively well at room temperature, to form quinones and muconic acids. Additionally, DIL was reduced to substances with a high phenolic-hydroxyl (OH) content (RDIL) under the presence of NaBH_4_, which further confirmed the composition of DIL and which can be applied to the development of lignin-based phenolic resins, providing a reference for the further modification as well as the utilization of DIL.

## 1. Introduction

Biomass is a naturally renewable organic-carbon resource, with great potential for replacing fossil fuels in the production of composite materials [1]. Lignocellulosic biomass is the most abundant form of biomass, of which lignin is one of the main components, accounting for 10–25% of the total lignocellulosic biomass [2]. Lignin, as a natural biopolymer, consists of three basic units, namely *p*-hydroxyphenyl (H), guaiacyl (G), and syringyl (S) units, through C-O-C and C-C linkages [3]. Its unique aromatic structure gives it the potential to serve as a reliable source for the production of polymeric materials, high-value fuels, bulk compounds and fine chemicals, reducing the use of fossil resources and contributing to the sustainable development of society. Industrial lignin is easy to obtain and cheap, and a large amount of lignin is produced in the production of bioethanol, pulp and paper [4,5], but lignin is not effectively utilized, due to its complex and diverse structure and low activity. Most lignin is used for energy combustion, resulting in a great waste of resources [6]. Noticeably, the degradation of lignin to obtain oligomers or small molecule compounds with reactive groups by derivatization, to prepare high value-added products, is one of the effective paths for utilizing lignin. Currently, most researchers degrade lignin under high temperature and high pressure, to obtain target products. Lavoie et al. treated softwood lignin with 5 wt.% sodium hydroxide solution at a temperature of 300–350 °C and a pressure of 9–13 MPa to obtain low-molecular products such as vanillin and 5-methylguaiacol [7]. Gasson et al. used formic acid as a hydrogen supply reagent to rapidly degrade lignin in ethanol under high temperature and pressure to produce syringol and guaiacol [8]. Mahmood et al. tackled lignin with a water-ethanol co-solvent for 1 h at 2 MPa and 250 °C, to obtain degradation products with *M*_w_ less than 1000 g·mol^−1^ [9]. Rana et al. employed water-ethanol-formic acid at 330 °C for 2 h to treat the concentrated acid hydrolysis of lignin to obtain monomeric compounds such as 4-propyl guaiacol, 4-methyl guaiacol and catechol [10]. Although lignin has been degraded relatively well under high temperature and pressure, the operations are complicated and the reaction conditions are strict. Hence, lignin degradation under mild conditions has been followed with interest.

ClO_2_ is a safe, efficient and green oxidant with an oxidation capacity 2.63 times higher than chlorine, and oxidizes with a variety of organic and inorganic substances without producing harmful substances; thus, it is widely used in water disinfection, medical disinfection, fruit, vegetable disinfection and pulp bleaching [11]. ClO_2_, as the main bleaching agent in pulp and paper, is characterized by the selective oxidation of lignin and pigments under mild conditions. Del Río et al. pointed out that the structural changes of lignin during the delignification of wood by the sulfate method mainly contained side-chain degradation, the cleavage of β-O-4 and limited demethylation [12]. Rencoret et al. found that Soda-AQ pulping preferentially removed S-lignin and fractured the β-O-4 linkages, and that the residual lignin was structurally similar to natural lignin [13]. Wang et al. elucidated the fact that ClO_2_ bleaching resulted in the significant degradation of lignin, which was also attributed to the reduction of the β-O-4 bond [14]. Therefore, compared with the use of high temperature and harsh reaction-conditions, the gentle degradation of lignin by ClO_2_ has great attraction.

At present, there are few studies on the oxidative degradation of directly isolated lignin with ClO_2,_ and the subsequent modification and utilization of degradation products has also not been reported. Consequently, the direct oxidation of lignin using ClO_2_ and the analysis of the structure of degradation products are of great importance for further modification and utilization of lignin. In this study, IL was degraded using a self-made ClO_2_ solution in acidic aqueous suspension, at room temperature. The effects of the technological conditions on IL degradation and structure were investigated. The reduction of DIL with NaBH_4_ is used to explore the composition of DIL, which also provides ideas for the modification of DIL. Gel permeation chromatography (GPC), Fourier-transform infrared spectroscopy (FT-IR), X-ray photoelectron spectroscopy (XPS) and nuclear magnetic resonance spectroscopy (^1^H NMR, ^31^P NMR and 2D-HSQC NMR) were used to study the structural changes of IL during degradation and reduction, and the evolution of the products. The possible mechanism of lignin degradation by ClO_2_ at room temperature was proposed, which laid the foundation for further research and the utilization of ClO_2_ oxidative-degradation products.

## 2. Results and Discussion

### 2.1. Optimization of ClO_2_ Degradation-Process Conditions

Lignin is by far the most abundant renewable aromatic-resource in nature, and its unique aromatic structure makes it a stable raw material for aromatic chemicals. When ClO_2_ oxidatively degrades IL, some of the aromatic structures of IL undergo ring opening. In order to minimize such phenomena and achieve an excellent degradation effect, the amount of ClO_2_ solution added, the reaction-system pH and the reaction time were optimized, to determine the optimal process condition. DILs represent degradation products under the exploration of optimal conditions.

The concentration of ClO_2_ determines its electron-donating ability, and significantly affects the efficiency of IL oxidative degradation. The functional groups of IL/DILs and their corresponding positions are shown in Table 1 [15]. As illustrated in Figure 1a, with an increase in the amount of ClO_2_ solution added, the peak intensity of the carbonyl (C=O) diffraction peak at 1700 cm^−1^ increased continuously, while the peak strengths of the benzene-ring characteristic peaks at 1596 m^−1^ and 1508 m^−1^ decreased continuously, indicating that the oxidative degradation of IL generated C=O-containing substances, and some aromatic rings opened. Compared to IL, the molecular weight of DILs first decreased and then increased with the addition of the ClO_2_ solution (Figure 1d). The reason is that the increase in the amount of ClO_2_ produces more radical intermediates (radical cation intermediates and phenoxyl radical intermediates), leading to the repolymerization reaction of the lignin fragments [16]. The molecular weight of the DIL (*M*_n_ = 2524 g·mol^−1^ and *M*_w_ = 3348 g·mol^−1^) was lowest when ClO_2_ was added at 30 mL.

Under acidic conditions, when the pH drops below 2.00, HClO generated from ClO_2_ is further converted to Cl_2_, which reacts with lignin to form adsorbable organic halogen (AOX) [17]. Therefore, the reaction-system pH is crucial for the degradation of IL. Figure 1b shows that with the increase in the reaction-system pH, the increase in the peak intensity of the C=O diffraction peak at 1700 cm^−1^ became smaller; the peak intensities of the characteristic peaks of the benzene ring at 1596 cm^−1^ and 1508 m^−1^ were lowest at pH 3.5, then gradually increased, and finally tended to be constant. Meanwhile, the molecular weight and PDI of DIL were minimized when the pH was 3.5, as seen in Figure 1e. In order to reduce the ring-opening phenomenon and achieve the degradation effect, pH 4.5 is the optimal choice for the degradation process.

Furthermore, high efficiency in a short time can save production costs in chemical reactions [18]. In Figure 1c,f, within the reaction time of 1–3 h, the molecular weight and PDI of DILs decreased with the increase in reaction time. The peak intensities of the characteristic peaks of the benzene ring at 1596 m^−1^ and 1508 m^−1^ exhibited the same trend, while the peak intensities of the characteristic diffraction peaks of the C=O group at 1700 cm^−1^ increased with the extension of the time. When the reaction time were 4 h and 5 h, ClO_2_ was completely consumed, the molecular weight and PDI of the DILs did not decrease, and the benzene ring no longer continued to open.

In summary, the optimal conditions for the oxidative degradation of IL by ClO_2_ in aqueous phase at room temperature are: 30 mL ClO_2_ solution (2.5 mg·L^−1^) and the reaction-system pH 4.5 and 3 h.

### 2.2. Structural Characterizations of IL before and after Degradation

The structure of DIL under the optimal conditions was characterized and compared with that of raw IL, to explore the evolution of substances in the degradation process.

#### 2.2.1. FT-IR Analysis

In order to quantitatively study the structural changes of IL in the degradation process, the structures of IL before and after degradation were characterized using FT-IR spectroscopy (Figure 2). When IL was degraded by ClO_2_, the relative intensities of the characteristic absorption peaks of the aromatic ring at 1596 cm^−1^ and 1508 cm^−1^ were feeble, indicating that the aromatic structure of IL appeared open after degradation. The absorption bands situated at 2842 cm^−1^ and 1454 cm^−1^ were the stretching vibration peaks of C-H and C-O in -OCH_3_, respectively, and the relative intensities of the peaks weakened after degradation, due to the demethylation reaction of the lignin during degradation. Liu et al. elaborated on the fact that the phenolic structure in lignin first generated phenoxyl radicals with ClO_2_, which continued to react with ClO_2_ to generate oxidation products [15], so the intensity of the phenolic OH absorption-peak at 1212 cm^−1^ was attenuated after degradation. In addition, the intensity of the ether bond absorption-peak at 1026 cm^−1^ also decreased. The β-O-4 bond has the lowest dissociation energy of 277.1 kJ·mol^−1^ among lignin bonds, and the β-O-4 bond is most susceptible to breakage, resulting in the lower molecular weight of lignin during the degradation reaction [19,20]; it is therefore presumed that the β-O-4 ether bond is the first targeted bond to break at room temperature during the degradation process. However, the absorption peak of the C=O group at 1700 cm^−1^ was significantly enhanced after degradation, meaning that the degradation of IL by ClO_2_ produced C=O-containing substances.

#### 2.2.2. ^1^H NMR Analysis

To deeply explore the structural changes of IL before and after degradation, especially the ring opening of the IL aromatic structure, a quantitative analysis of the important functional groups of IL before and after degradation was carried out using ^1^H NMR spectroscopy [21]. Figure 3a,b display the ^1^H NMR spectra before and after IL degradation. Table 2 summarizes the chemical shifts and quantitative-analysis results of the functional groups of IL and DIL [22,23]. The ^1^H NMR spectra of IL and DIL were similar, but the quantitative results of the functional groups were different. The broad-peak signals at 7.84–6.10 ppm belonged to aromatic protons. Among them, the spectral bands at 7.78–7.41 ppm were associated with aromatic protons in the *p*-hydroxyphenyl (H) structure, the broad-peak range from 7.41–6.85 ppm was attributed to aromatic protons in the guaiacyl (G) structure, the specific signals from 6.85 to 6.10 ppm represented the syringyl (S) aromatic protons, with the contents of 0.58 mmol·g^−1^, 1.80 mmol·g^−1^ and 3.27 mmol·g^−1^, respectively, and the total amount was 6.10 mmol·g^−1^. After degradation, the content of each aromatic proton decreased to 0.48 mmol·g^−1^, 1.58 mmol·g^−1^ and 2.69 mmol·g^−1^, respectively, and the total content decreased to 4.75 mmol·g^−1^, manifesting that the IL partial aromatic-ring was opened in the degradation process, which confirmed the results of the FT-IR spectrum. The ^1^H NMR spectrum showed that the β-O-4 bond was located at 4.93 ppm, and the β-O-4 bond contents of IL and DIL were 1.05 mmol·g^−1^ and 0.74 mmol·g^−1^, respectively, indicating that the β-O-4 bond was broken during the degradation process, which would lead to a decrease in the molecular weight of DIL. While the broad spectral region in the range of 9.6–8.9 ppm was detected, the amount of the phenolic-OH group of DIL (2.22 mmol·g^−1^) was lower than that of the IL (2.43 mmol·g^−1^), indicating that the degradation of IL by ClO_2_ was a process that consumed phenolic OH. However, the amount of phenolic OH did not decrease drastically during the degradation process, because the breakage of the β-O-4 bond generated a new phenolic OH group.

#### 2.2.3. 2D-HSQC Analysis

Two-dimensional-HSQC was utilized to further confirm the existence of the β-O-4 bond fracture in IL after ClO_2_ treatment. The 2D-HSQC spectra of IL and DIL can be divided into three regions, namely the aliphatic regions, side-chain regions and aromatic-ring regions. Since the aliphatic group cannot provide more information on the composition and structure of lignin, it is rarely analyzed and discussed. The side-chain (δ_C_/δ_H_ 50–90/2.7–6.0 ppm) and aromatic-ring (δ_C_/δ_H_ 100–150/5.5–8.5 ppm) regions of IL and DIL are shown in Figure 3c,d. In the side-chain regions of IL and DIL, -OCH_3_ group could be observed at δ_C_/δ_H_ 56.0/3.74 in the spectra. The obvious signals at δ_C_/δ_H_ 59.5/3.35–3.80 ppm corresponded to C_γ_-H_γ_ correlation in the β-O-4 aryl ether (A) structure. The C_γ_-H_γ_ associated signal in the phenylcoumaran (C) substructure was confirmed by the appearance of a peak at δ_C_/δ_H_ 62.3/3.70 ppm. The signal of the α-position in the side chain of the β-O-4 linkages correlation was found at δ_C_/δ_H_ 71.6/4.83 ppm, whereas the signal of the β-position in the side chain linked to the S unit in the β-O-4 linkages was observed at δ_C_/δ_H_ 85.9/4.11. Moreover, the obvious signal at δ_C_/δ_H_ 84.1/4.45 ppm corresponded to the C_β_-H_β_ in β-O-4 substructures (A) linked to a G unit in the side-chain regions of IL [24,25,26]. However, the signal disappeared in the side-chain region of DIL, suggesting that the β-O-4 bond broke after degradation. In the aromatic regions, the C_2,6_–H_2,6_ correlation of the S units was distinguished at δ_C_/δ_H_ 103.8/6.68 ppm; a minor but clear signal at δ_C_/δ_H_ 110.9/7.01 was related to the C_2_–H_2_ correlation of the G units. In addition, a correlated signal of the C_2,6_–H_2,6_ correlation from the H units could be detected at δ_C_/δ_H_ 128.0/7.10 [27]. The linkage-bond content was obtained by the quantitative calculation of the β-O-4 linkage bond in IL and DIL. The content of the β-O-4 of IL and DIL was 67.86% and 72.65% respectively. As the content of the aromatic rings decreased after ClO_2_ treatment, the content of the β-O-4 of DIL was higher than IL, explaining that the β-O-4 in IL was fractured.

#### 2.2.4. XPS Analysis

Figure 4 demonstrates the fact that IL and DIL are mainly composed of C and O elements. To further understand the changes in the chemical structure and relative content of IL before and after degradation, a split-peak fit was performed for the C and O elements. In the C1s spectrum (Figure 5(a_1_,a_2_)), the spectrum can be divided into four peaks at 284.8, 286.2, 287.3, and 288.2 eV, corresponding to C-H/C-C/C=C, C-O-C, C=O, and -COO^−^, respectively [28]. Moreover, the presence of three peaks, C=O (531.7 eV), C-O-C (532.96 eV) and C-OH (533.7 eV), was observed in the O1s spectrum (Figure 5(b_1_,b_2_)) [29,30]. The relative content of each chemical bond was calculated according to the sensitivity-factor method, and the results are shown in Table 3. Compared with IL, the content of the OH group, the ether bond or the -OCH_3_ content of DIL decreased, while the content of the C=O and carboxyl (-COOH) group increased, implying that the OH group of IL reacted to form muconic acid and quinones during the degradation process. Additionally, ^1^H NMR analysis found a change in the content of the β-O-4 bond, of −29.5%, while the XPS results exhibited a change in the relative content of the ether bond or -OCH_3_ of −34.9%, inferring that the content of -OCH_3_ was decreased during degradation. Therefore, the demethylation reaction occurred during the degradation of IL.

#### 2.2.5. ^31^P NMR Analysis

^31^P NMR is an advanced technology to distinguish the types of OH group in samples, and is used to determine the content of OH and -COOH [31]. The content of aliphatic OH, phenolic OH, and the -COOH of IL and DIL were determined using ^31^P NMR technology to further clarify the changes in functional-group content and the formation of substances during IL degradation. Figure 6a,b manifest the ^31^P NMR spectra of IL and DIL, respectively. The ^31^P NMR signal-distribution and corresponding quantification results of the different OH and -COOH of IL and DIL are demonstrated in Table 4. In Figure 6a,b, the specific quantization intervals are as follows: the aliphatic OH is related to 149.7–145.5 ppm, the syringyl (S) phenolic OH and condensed phenolic OH are located at 144.3–141.1 ppm, the guaiacyl (G) phenolic OH is represented at 140.7–138.5 ppm, and the *p*-hydroxyphenyl (H) phenolic OH is found at 138.3–136.4 ppm. In addition, the signal appearing at 136.0–133.1 ppm belongs to the -COOH group [32]. The contents of the S-OH and condensed phenolic OH, G-OH and H-OH in IL were 0.82 mmol·g^−1^, 1.01 mmol·g^−1^ and 0.59 mmol·g^−1^, respectively. After degradation by ClO_2_, the contents of the S-OH and condensed phenolic OH, G-OH and H-OH decreased to 0.77 mmol·g^−1^, 0.97 mmol·g^−1^ and 0.51 mmol·g^−1^, respectively, while the content of the -COOH group of DIL (1.05 mmol·g^−1^) was higher than IL (0.94 mmol·g^−1^), indicating the production of muconic acid during degradation, which was consistent with the semi-quantitative results of the XPS. The total phenolic OH group of IL was 2.42 mmol·g^−1^ and the total phenolic OH group of DIL was 2.25 mmol·g^−1^, which corroborated with the content of the phenolic OH group in the quantitative ^1^H NMR spectrum.

#### 2.2.6. GPC Analysis

Figure 6c reveals the GPC curves, *M*_w_, *M*_n_ and PDI of IL and DIL. Compared with the GPC curve of IL, the GPC curve of DIL shifted significantly toward the low molecular weight. Furthermore, the *M*_w_ and *M*_n_ of DIL were lower than the *M*_w_ and *M*_n_ of IL, suggesting a significant degradation effect of ClO_2_ on IL. The ^1^H NMR analysis displayed a decrease in the β-O-4 bond from 1.05 mmol·g^−1^ to 0.74 mmol·g^−1^. Notably, the 2D-HSQC results also showed the presence of β-O-4 breakage during oxidation. Therefore, the decrease of *M*_w_ and *M*_n_ of DIL was attributed to the cleavage of the β-O-4 bond. In addition, the PDI of DIL was lower than that of IL, explaining the fact that the molecular-weight distribution of DIL became narrower and more uniform.

### 2.3. Reduction Structural Analysis of DIL

#### 2.3.1. FT-IR Analysis

NaBH_4_ can reduce C=O compounds including quinones and aromatic ketones [33]. DIL is reduced by NaBH_4_, and the structures of DIL and RDIL are characterized using FT-IR spectroscopy, as shown in Figure 6d. Compared with DIL, the relative intensities of the characteristic absorption-peaks of RDIL aromatic structures at 1596 cm^−1^ and 1508 cm^−1^ r and the relative intensity of the phenolic OH peak located at 1212 cm^−1^ increased, while the intensity of the C=O absorption peak at 1700 cm^−1^ weakened. The aromatic structure of IL opened the ring to form muconic acid after degradation, so the relative intensities of the absorption peaks of the aromatic ring of RDIL were lower than IL. The results indicated that ClO_2_ degraded IL to form quinones, which were converted to a phenol-type substance after reduction by NaBH_4_, increasing the phenolic OH content of DIL.

#### 2.3.2. ^1^H NMR Analysis

The aromatic rings of the product before and after reduction were quantitatively analyzed using ^1^H-NMR (Figure 7a,b). The spectral bands of 7.78–7.41 ppm, 7.41–6.85 ppm and 7.84–6.20 ppm corresponded to the aromatic protons of *p*-hydroxyphenyl, guaiacyl and syringyl, respectively, and their contents after reduction were 0.53 mmol·g^−1^, 1.76 mmol·g^−1^ and 3.06 mmol·g^−1^, respectively. The total content was 5.35 mmol·g^−1^, which was 0.60 mmol·g^−1^ higher than the total aromatic protons of DIL. Meanwhile, the content of the phenolic proton in the broad-spectrum region in the range of 9.21–9.98 ppm increased from 2.22 mmol·g^−1^ to 2.74 mmol·g^−1^ by reduction. Thus, the quinones produced during IL degradation were selectively reduced to phenol substances under the action of NaBH_4_, providing a basis for the further modification of DIL.

### 2.4. Possible Degradation Mechanism

The possible reaction mechanism of IL degradation by ClO_2_ under acidic conditions in the aqueous phase at room temperature, is as follows (Figure 7c) [34,35,36]. The initial steps in the degradation of IL by ClO_2_ include the ClO_2_ electrophilic addition to the aromatic ring in the non-phenolic structure undergoing a single-electron transfer to form radical cation intermediates under acidic conditions in which chlorite and hypochlorite are generated or phenoxy radical intermediates are formed, with the phenolic structure. The radical cation intermediates and the phenoxy radical intermediates were further attacked by ClO_2_ to form an unstable chlorite structure, which were decomposed by demethylation and hydrolysis reactions to form quinones and muconic acid. In addition, the β-O-4 linkage bond of IL was partially broken in the presence of ClO_2_.

## 3. Materials and Methods

### 3.1. Materials

The IL was sourced from Shandong Longli Co., Ltd. (Shandong, China). Sodium chlorite (NaClO_2_), NaBH_4_ and sodium hydroxide (NaOH), chemically pure, were all from Shanghai Macklin Biochemical Co., Ltd. (Shanghai, China). Methanol and sulfuric acid (H_2_SO_4_) were analytical pure reagents and all provided by the Nanjing Chemical Reagent Co., Ltd. (Nanjing, China). Sodium thiosulfate (Na_2_S_2_O_3_) standard solution (0.1000 mol·L^−1^), analytically pure, was purchased from the Shanghai Aladdin Reagent Co., Ltd. (Shanghai, China). Potassium phosphate monobasic (KH_2_PO_4_), sodium phosphate dibasic dodecahydrate (Na_2_HPO_4_·12H_2_O), potassium iodide (KI) and soluble starch, were chemically pure, and were all from the Shanghai Aladdin Reagent Co., Ltd. (Shanghai, China).

### 3.2. Preparation of ClO_2_ Solution

ClO_2_ gas was generated using 10 wt.% NaClO_2_ solution and 20 wt.% H_2_SO_4_ solution, and ClO_2_ solution was successfully prepared by absorbing the gas with pure water at 0 °C. The reaction equation is shown in Equation (1) [37]. The concentration of the prepared ClO_2_ solution was determined according to the iodometric titration method [38].
5NaClO_2_ + 2H_2_SO_4_ = 4ClO_2_ + 2Na_2_SO_4_ + NaCl + 2H_2_O(1)

### 3.3. Degradation of Lignin by ClO_2_

A 250 mL three-necked flask with a stirring bar and a constant pressure funnel was filled with 1 g IL and 40 mL pure water, and stirred thoroughly. The suspension pH was adjusted with 20 wt.% NaOH solution, and then self-made ClO_2_ solution was put into the suspension. After the ClO_2_ solution had been completely added, it was allowed to stir at room temperature for a period of time. The acquired DIL was separated by filtration, washed several times with distilled water, and dried under vacuum at 45 °C for 12 h. The whole reaction process needed to be carried out in the dark.

### 3.4. Reduction of DIL by NaBH_4_

A total of 1 g DIL was taken in a 250 mL clean beaker and added to 50 mL pure water; 20 wt.% NaOH solution was added dropwise to the beaker with stirring to dissolve the DIL. After the DIL dissolution, 100 mL methanol was added to the solution and stirred for 10 min, and 0.1 g NaBH_4_ was then added, with stirring for 12 h at room temperature. The reaction was quenched with pure water. All the methanol and part of the pure water were then spun out, and the concentrate pH was adjusted to 2 to precipitate the solid, the solid was filtered and washed with pure water several times, and finally dried under vacuum at 45 °C for 12 h.

### 3.5. Characterizations

The chemical structure of IL/DIL/RDIL was characterized using a Nicolet IS10 Fourier-transform infrared spectrometer (Nicolet company, Madison, WI, USA) in the range of 500–4000 cm^−1^, with a resolution of 4 cm^−1^.

A total of 25 mg IL/DIL/RDIL and 0.044 mmol 1,3,5-trioxane formaldehyde (internal standard) were dissolved in 0.5 mL of DMSO-d_6_ (solvent). The ^1^H NMR spectra of the samples were recorded on a Bruker 400 spectrometer (Bruker, Karlsruhe, Germany) with 128 scans. The content of the functional group was calculated according to the integral ratio of the proton of the functional group to the internal standard proton [39].

The 2D-HSQC NMR analysis method: 50 mg IL/DIL was dissolved in 0.5 mL DMSO-d_6_ and then transferred to NMR tubes for analysis at 25 °C. The ^1^H and ^13^C spectral widths were 5000 and 20,000 Hz, respectively. The 2D-HSQC spectra were collected and recorded using a Bruker AVIII 400 MHz spectrometer, and the data were processed on the corresponding software. The solvent peak DMSO-d_6_ (δ_C_/δ_H_ 39.5/2.49 ppm) was used as the internal standard peak for chemical-shift correction [40]. A semi-quantitative analysis of relevant connecting bonds in the 2D-HSQC spectrum of IL/DIL was carried out, according to the following formula [41]:I(C_9_) units = 0.5 I(S_2,6_) + I(G_2_) + 0.5 I(H_2,6_)(2)
I(X)% = I(X)/I(C_9_) × 100%(3)

I(S_2,6_), I(G_2_), I(H_2,6_) and I(X) represent the integral value of the S_2,6_ signal, the integral value of the G_2_ signal, the integral value of the H_2,6_ signal and the integral value of each connecting bond in IL/DIL, respectively.

The X-ray source of the EscaLab Xi+ X-ray photoelectron spectrometer (Mercy Fisher Scientific, Waltham, MA, USA) was AIKa, and the full-spectrum and fine-spectrum data were obtained using coarse scanning and narrow scanning.
(4)$$C_{i}=\frac{A_{i}/S_{j}}{\sum_{i}^{j}A_{\mathrm{ij}}/S_{i}}$$

The relative atomic concentration (C_i_) of the functional group was calculated according to Formula (4) [42]. A—peak area of spectral peak, S—relative-sensitivity factor.

The pyridine/deuterated chloroform solution (solution A) was obtained by mixing a specific amount of analysis pure pyridine and deuterated chloroform (*v*/*v* = 1.6:1). A total of 54.25 mg cyclohexanol was accurately weighed, and the volume was fixed to 5 mL with solution A, to obtain the required internal standard solution (solution B). A total of 25 mg chromium (III) acetylacetonate was weighed, and the volume was fixed to 5 mL with solution A, to obtain 5 mg·mL^−1^ relaxation reagent solution (solution C). A total of 20 mg IL/DIL was accurately weighed, and then 500 μL solution A, 100 μL solution B and 100 μL solution C were added, fully mixed, and 100 μL 2-chloro-4,4,5,5-tetracmethyl-1,3,2-dioxaphospholane (TMDP) was added for phosphating treatment, and determined after 15 min [43]. The ^31^P NMR spectra were acquired on a Bruker-Advance III 400 MHz (Bruker, Karlsruhe, Germany) NMR instrument, with a pulse angle of 30°, relaxation delay (d_1_) of 2 s, 64 K data points, and 2048 scans.

A total of 2 g IL/DIL was added to a pyridine−acetic anhydride solution (*v*/*v* = 1:1, 30 mL), stirred thoroughly, and placed in the dark for 48 h at room temperature. The target was then obtained by adding 1% HCl under ice water, washing the target several times with distilled water to neutral, and drying under vacuum to obtain acetylated lignin [44]. A total of 5 mg of the acetylated sample was dissolved in 1 mL tetrahydrofuran (THF), and then 50μL of the solution was injected into a high-performance gel chromatograph (waters, Milford, MA, USA) for determination, with a flow rate of 1 mL/min and a column temperature of 30 °C.

## 4. Conclusions

ClO_2_ is capable of degrading lignin under mild conditions through the partial ring-opening of the lignin aromatic ring and the cleavage of β-O-4. In this mild degradation process, the content of the β-O-4 bond decreased from 1.05 mmol·g^−1^ to 0.74 mmol·g^−1^, and the *M*_n_ of IL decreased from 4130 g·mol^−1^ to 2618 g·mol^−1^, The *M*_w_ decreased from 6563 g·mol^−1^ to 3831 g·mol^−1^. Concurrently, the aromatic-proton content in IL decreased from 6.10 mmol·g^−1^ to 4.75 mmol·g^−1^, and increased to 5.35 mmol·g^−1^ with NaBH4, the phenolic hydroxyl (OH) group decreased from 2.43 mmol·g^−1^ to 2.22 mmol·g^−1^, and increased to 2.74 mmol·g^−1^ after reduction, indicating that quinones were generated during the degradation of IL, which was finally reduced to RDIL, RDIL can replace part of the phenol, to produce phenolic resin. Moreover, ^31^P NMR results suggested that the oxidation products obtained muconic acid with a yield of 11.70%; this can be used as excellent flame-resistance material or as feedstocks for chemical production. In short, the proposed lignin depolymerization process will be a beneficial reference for the future utilization of this sustained and aromatic material, which is a potential energy source. In addition, RDIL was obtained by using NaBH_4_, which indicated the feasibility of DIL modification.

## Figures and Tables

**Figure 1 ijms-24-01479-f001:**
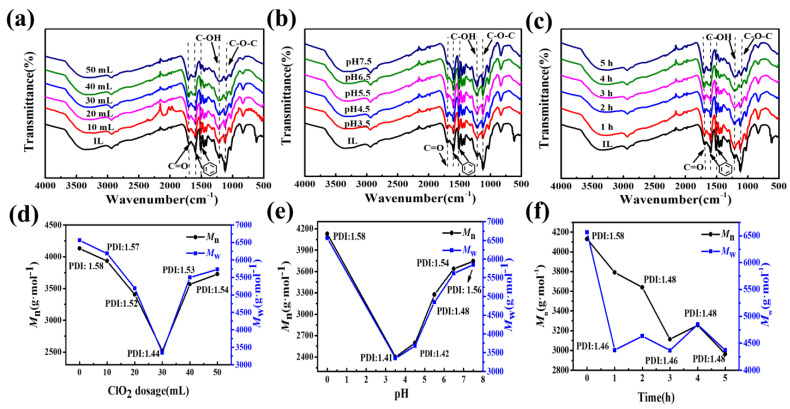
Effect of ClO_2_ dosage: (**a**) reaction system pH and (**b**) reaction time (**c**) on IL degradation, and effect of ClO_2_ dosage: (**d**) reaction system pH and (**e**) reaction time (**f**) on IL molecular weight. Conditions: (**a**) 10, 20, 30, 40 and 50 mL 2.5 mg·L^−1^ ClO_2_, pH 4.5, 5 h; (**b**) 30 mL 2.5 mg·L^−1^ ClO_2_, pH 3.5, 4.5, 5.5, 6.5 and 7.5, 5 h; (**c**) 30 mL 2.5 mg·L^−1^ ClO_2_, pH 4.5, 1 h, 2 h, 3 h, 4 h and 5 h.

**Figure 2 ijms-24-01479-f002:**
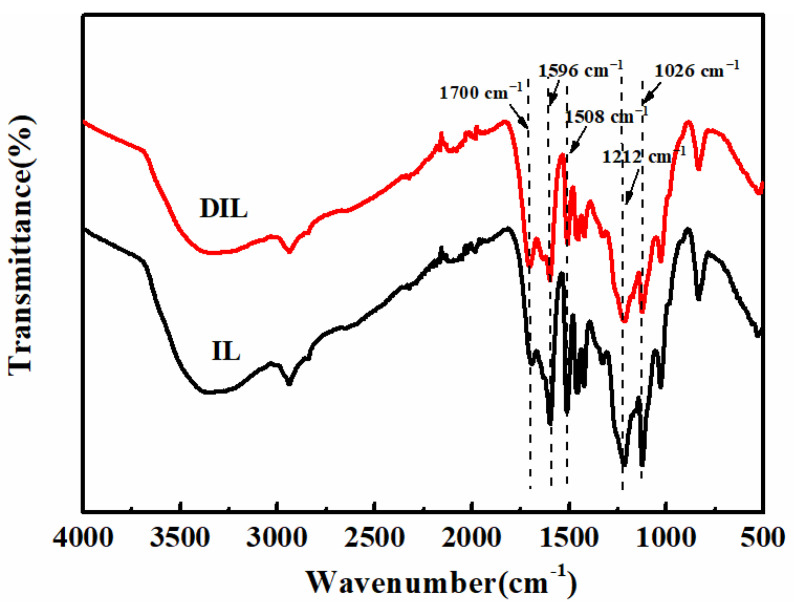
FT-IR spectra of IL and DIL.

**Figure 3 ijms-24-01479-f003:**
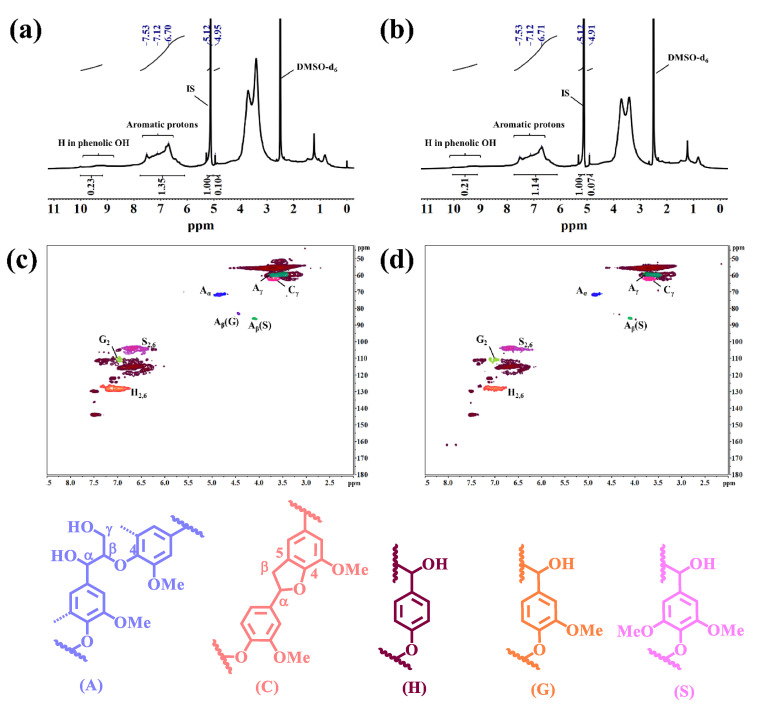
^1^H NMR spectra of IL (**a**) and DIL (**b**), 2D-HSQC spectra of IL (**c**) and DIL (**d**), and (A) β-O-4′ aryl ether linkages with a free-OH at the γ-carbon; (C) phenylcoumarane substructures formed by β-5′ and α-O-4′ linkages; (H) *p*-hydroxyphenyl units; (G) guaiacyl units; (S) syringyl units.

**Figure 4 ijms-24-01479-f004:**
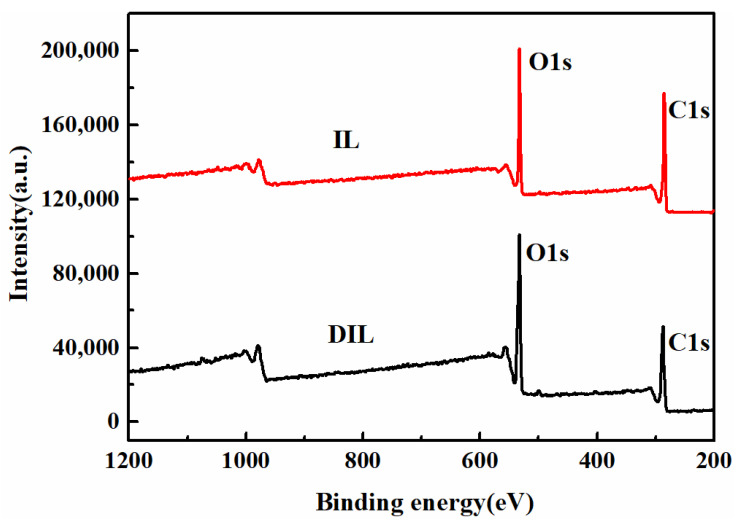
XPS spectrum of IL and DIL.

**Figure 5 ijms-24-01479-f005:**
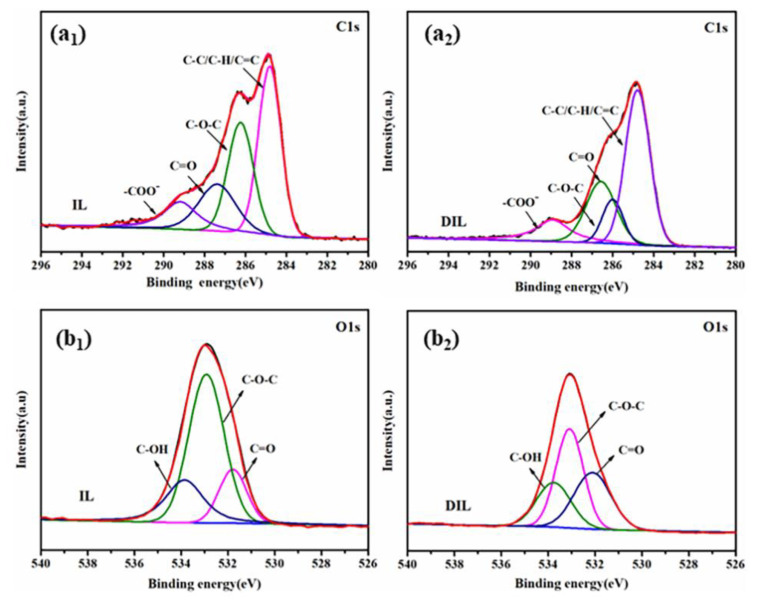
C1s (**a_1_**,**a_2_**) and O1s (**b_1_**,**b_2_**) spectra for IL and DIL.

**Figure 6 ijms-24-01479-f006:**
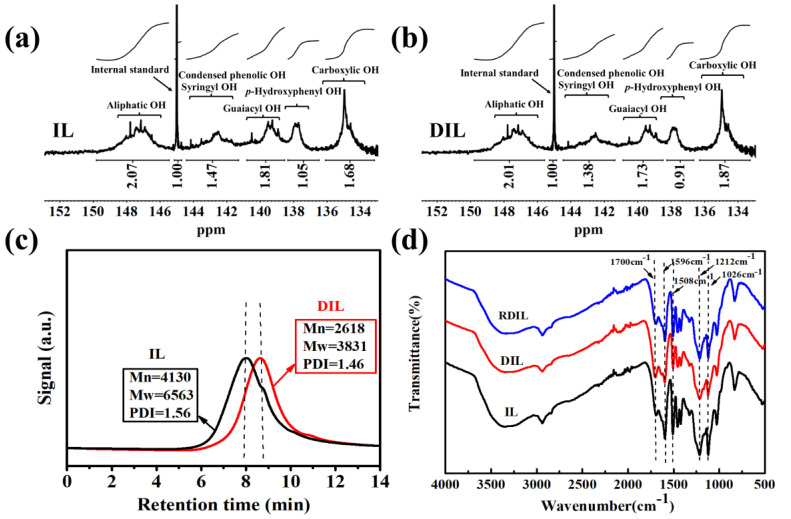
^31^P NMR spectra of IL (**a**) and DIL (**b**), GPC spectra of IL and DIL (**c**), and FT-IR spectra of IL, DIL and RDIL (**d**).

**Figure 7 ijms-24-01479-f007:**
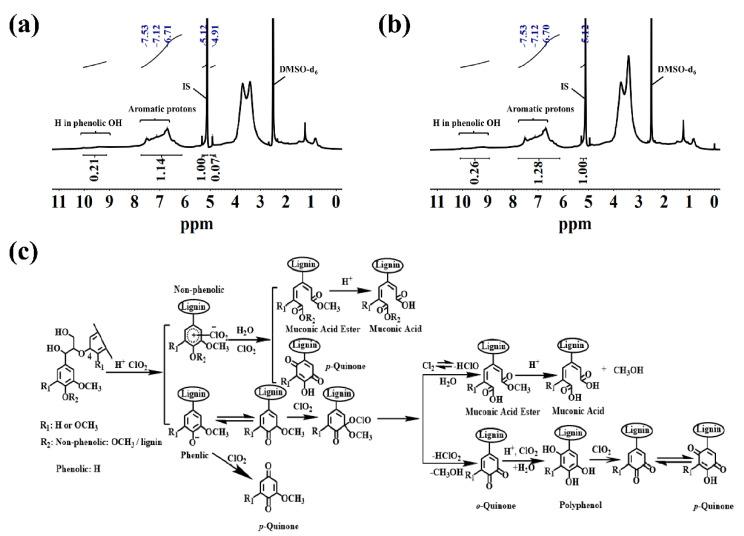
^1^H NMR spectra of DIL (**a**) and RDIL (**b**), and possible mechanism of IL degradation by ClO_2_ (**c**).

**Table 1 ijms-24-01479-t001:** Functional groups of IL/DILs and their corresponding positions.

Functional Groups	Wavenumber (cm^−1^)
-OH	3253
-CH_2_, -CH_3_	2934
C-H in -OCH_3_	2840
C=O	1674
Aromatic ring vibrations	1511
	1592
C-H in -OCH_3_	1451
C-O in phenolic OH	1211
Ether bond (C-O-C)	1030
C-H in p-hydroxyphenyl units	816

**Table 2 ijms-24-01479-t002:** Quantitative analysis of IL and DIL using ^1^H NMR.

Chemical Shift (ppm)	Functional Groups	IL (mmol·g^−1^)	DIL (mmol·g^−1^)
2.54	DMSO-d_6_	-	-
4.93	β-O-4 bond	1.05	0.74
5.12	1,3,5-trioxane	-	-
7.78–7.41	Aromatic H in H unit	0.58	0.48
7.41–6.85	Aromatic H in G unit	1.80	1.58
6.85–6.10	Aromatic H in S unit	3.27	2.69
9.21–9.98	Proton in phenolic OH	2.43	2.22

**Table 3 ijms-24-01479-t003:** Relative changes of characteristic-groups content of IL before and after degradation.

Characteristic Groups	Relative Content Change
C-OH	−14.3%
C-O-C or -OCH_3_	−34.9%
C=O	+16.5%
-COOH	+12.0%

**Table 4 ijms-24-01479-t004:** Quantitative analysis of ^31^P NMR signal distribution and chemical structure of IL and DIL.

Chemical Shift (ppm)	Functional Groups	IL (mmol·g^−1^)	DIL (mmol·g^−1^)
149.7–145.5	Aliphatic OH	1.16	1.13
145.1–144.7	Cyclohexanol	-	-
144.3–141.1	S-OH and condensed phenolic OH	0.82	0.77
140.7–138.5	G-OH	1.01	0.97
138.3–136.4	H-OH	0.59	0.51
136.0–133.1	-COOH	0.94	1.05

## Data Availability

The data presented in this study are available in the manuscript’s figure.

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
