# Peer review of "Lignin Degradation via Chlorine Dioxide at Room Temperature: Chemical Groups and Structural Characterization"

_ijms, 2023, doi:10.3390/ijms24021479_

Round 1

Reviewer 1 Report

The submitted manuscript deals with the oxidative degradation of lignin by chlorine dioxide. Chlorine dioxide is a reagent that has been widely used for the bleaching of paper pulp, as well as a reagent for the analysis of pulps by oxidation of lignin and obtaining holocellulose. In this manuscript, the study of the oxidative degradation of lignin is proposed, but I do not quite understand what the purpose of this study is. Is it intended to obtain some relevant or interesting oxidation product? Such as vanillin, which is a product of the lignin oxidation, simple phenols with good reactivity?, or just determine the reaction mechanism. In my opinion, the manuscript needs a lot of work to be published, and to define what its objectives are, and for that reason my recommendation is for a major revision. The title "Chemical groups and structural characterization of lignin via chlorine dioxide-mediated degradation at room temperature" suggests the structural lack of lignin, and from my point of view this is not accurate. Degradation by chlorine dioxide is not used for the characterization, but the products obtained from the oxidation are characterized and compared with the initial ones.    

Reviewer 2 Report

This work used chlorine dioxide for the oxidative degradation of lignin and provided  sufficient data. Therefore, I am pleased to recommend publication after the authors have resolved the following issues:

1. In the introduction, the authors may wish to emphasize the meaning of work and the bottleneck in oxidative degradation of lignin using chlorine dioxide.

2. For XPS analysis, the elemental compositions of IL and DIL should be provided. Furthermore, the actual content of characteristic groups before and after lignin degradation should be included in Table 3.

3. Can the authors determine the lignin degradation rate?

4. Please check the ms carefully. Some mistakes should be corrected. For example, the formula 2-2.

Round 2

Reviewer 1 Report

In this new version it is clarified what is the purpose of the work, mainly whit the modifications made.   I don't like the new title of the manuscript and I suggest something similar to "Lignin degradation by chlorine dioxide at room temperature: chemical groups and structural characterization".
